# Analysis of genetically independent phenotypes identifies shared genetic factors associated with chronic musculoskeletal pain conditions

Yakov A. Tsepilov [1,2,12], Maxim B. Freidin[3,12], Alexandra S. Shadrina [1,2], Sodbo Z. Sharapov [1,2], Elizaveta E. Elgaeva [2], Jan van Zundert [4,5], Lennart C. Karssen [6], Pradeep Suri [7,8,9,10], Frances M. K. Williams [3] & Yurii S. Aulchenko [1,2,6,11✉]

Chronic musculoskeletal pain affects all aspects of human life. However, mechanisms of its genetic control remain poorly understood. Genetic studies of pain are complicated by the high complexity and heterogeneity of pain phenotypes. Here, we apply principal component analysis to reduce phenotype heterogeneity of chronic musculoskeletal pain at four locations: the back, neck/shoulder, hip, and knee. Using matrices of genetic covariances, we constructed four genetically independent phenotypes (GIPs) with the leading GIP (GIP1) explaining 78.4% of the genetic variance of the analyzed conditions, and GIP2–4 explain progressively less. We identified and replicated five GIP1-associated loci and one GIP2-associated locus and prioritized the most likely causal genes. For GIP1, we showed enrichment with multiple nervous system-related terms and genetic correlations with anthropometric, sociodemographic, psychiatric/personality traits and osteoarthritis. We suggest that GIP1 represents a biopsychological component of chronic musculoskeletal pain, related to physiological and psychological aspects and reflecting pain perception and processing.

[1] Laboratory of Theoretical and Applied Functional Genomics, Novosibirsk State University, 1 Pirogova Street, Novosibirsk, Russia 630090. [2] Laboratory of Recombination and Segregation Analysis, Institute of Cytology and Genetics, 10 Lavrentiev Avenue, Novosibirsk, Russia 630090. [3] Department of Twin Research and Genetic Epidemiology, School of Life Course Sciences, King's College London, London SE1 7EH, UK. [4] Department of Anesthesiology and Pain Medicine, Maastricht University Medical Centre, Maastricht, The Netherlands. [5] Department of Anesthesiology and Multidisciplinary Paincentre, ZOL Genk/Lanaken, Belgium. [6] PolyOmica, 5237 PA 's-Hertogenbosch, The Netherlands. [7] Division of Rehabilitation Care Services, VA Puget Sound Health Care System, 1660 S. Columbian Way, Seattle, WA 98108, USA. [8] Seattle Epidemiologic Research and Information Center (ERIC), Department of Veterans Affairs Office of Research and Development, 1660 S. Columbian way, Seattle, WA 98108, USA. [9] Department of Rehabilitation Medicine, University of Washington, 325 9th Ave, Seattle, WA 98104, USA. [10] Clinical Learning, Evidence, and Research (CLEAR) Center, University of Washington, 4333 Brooklyn Ave NEBox 359455Seattle, WA 98195, USA. [11] Department of Complex Genetics, Care and Public Health Research Institute, Maastricht University, Maastricht, The Netherlands. [12] These authors contributed equally: Yakov. A. Tsepilov, Maxim B. Freidin. ✉email: y.s.aulchenko@polyomica.com

Chronic pain is one of the most prevalent human health problems, affecting on average 20–30% of adults[1–3], and it is one of the most challenging conditions for clinical management[4]. Often chronic pain is present without a clear pathophysiological cause such as tissue damage and cannot be attributed to a known disorder. Chronic musculoskeletal pain is the most prevalent type of chronic pain in older adults[5]. Prevalence estimates vary widely depending on the studied population and the definition used to define these conditions[6]. For instance, in a Swedish study, the prevalence was 23.9% for chronic regional musculoskeletal pain and 11.4% for chronic widespread pain[7]. In Japan, the prevalence of chronic musculoskeletal pain was found to be 15.4% in the general population and reached 18.6% among individuals aged 40–49[8]. The most prevalent self-reported chronic musculoskeletal pain conditions are low back, neck, and shoulder pain[7,8]. According to the Global Burden of Disease Study 2015, low back pain and neck pain were the leading causes of global years lived with disability in 1990–2015[9].

Precise biological mechanisms underlying chronic pain are yet to be elucidated[10,11]. There is good evidence that chronic pain disorders are complex heritable traits[12,13]. Exploring the genetic underpinning of chronic pain phenotypes can expand basic knowledge on their etiology and biological mechanisms, improve diagnostics, and facilitate the development of effective therapies via the identification of therapeutic targets.

Genetic association studies have suggested a number of genes associated with chronic musculoskeletal pain phenotypes[14–16]. These studies were predominantly hypothesis-driven candidate-gene studies, which often had small samples sizes, and with some leading to conflicting results as has been borne out in other traits[17]. Compared to candidate-gene studies, genome-wide association studies (GWAS) offer an agnostic data-driven approach that allows identification of susceptibility genes without a prior mechanistic hypothesis. So far, only a few GWAS for forms of chronic musculoskeletal pain have been published, including chronic widespread pain[18], fibromyalgia[19], chronic back pain[20], sciatica[21], and painful temporomandibular disorder[22]. Thus, the genetic architecture of chronic musculoskeletal pain is far from being defined.

Research in chronic pain genetics faces a number of obstacles. According to the biopsychosocial model of pain, chronic pain results from a complex and dynamic interaction among biological, psychologic, and social factors[23]. The extreme complexity and heterogeneity of chronic pain phenotypes complicates identification of novel loci and makes it difficult to distinguish whether identified variants affect the risk of the primary pain-causing pathology (if any) or influence the development and maintenance of the chronic pain state itself. Both the primary underlying condition and its treatment, and the treatment of chronic pain, may confound studies. A study exemplifying these challenges is our recent GWAS of chronic back pain[20]. Despite the large sample size of nearly 158,000 individuals in the discovery sample and 284,000 subjects in the replication sample, we were able to detect and replicate only one locus. Thus, new strategies are required to improve understanding of the genetic influences in chronic pain conditions.

One possible solution to the problem of clinical heterogeneity is to study endophenotypes and subgroups of patients having different characteristics[15]. A complementary approach to reducing heterogeneity is to elucidate the common pathways shared by distinct pain phenotypes. Indeed, different chronic pain conditions may have common biological pathways such as those related to pain perception and processing. Several studies have provided evidence for shared genetic factors between conditions manifesting chronic pain[24] as well as pain at different anatomical sites[25,26]. However, to the best of our knowledge, no study yet published has explicitly identified these genetic factors.

Here, we investigated the genetic factors underlying chronic musculoskeletal pain reported at four locations (back, neck/shoulder, hip, and knee). These anatomical sites are commonly affected by osteoarthritis (OA). Pain is the predominant symptom of OA, but its intensity may be poorly correlated with OA severity based on pathological changes revealed by radiographs. Current evidence suggests that not only structural lesions, but also neuronal pathways and alterations of pain processing contribute to maintaining pain in OA patients[27]. We assumed that studying pain at multiple sites can unravel shared musculoskeletal pathways and, more importantly, provide deeper understanding of general chronic pain mechanisms. We used, to our knowledge, a novel approach to explore the genetic background of pain traits by analyzing genetically independent phenotypes (GIPs). Using data from UK Biobank[28] we identified and replicated specific loci associated with these GIPs, followed by in silico functional analysis, including a search for pleiotropic effects of functional variants, prioritization of likely causal genes, analysis of gene set and tissue enrichment, and estimation of genetic correlations with other complex traits.

## Results

**Overview of the study design**. Our study was designed to investigate the genetic components underlying chronic musculoskeletal pain at four locations: back, neck/shoulder, hip, and knee (Fig. 1). Individuals who reported more than 3 months of pain all over the body were not included in the present study. All studied pain phenotypes were found to have statistically significant SNP-based heritability (2–4% on the observed scale estimated by LD Score regression, and 5–7% on the observed scale estimated by REML algorithm, 7–9% on the liability scale estimated by LD Score regression, and 13–16% on the liability scale estimated by REML algorithm, Supplementary Data 1a) and to be genetically correlated with each other (Fig. 2c). Coefficients of phenotypic correlations between the studied pain traits ranged from 0.18 to 0.28 (Fig. 2d).

Using the matrix of genetic covariances between the studied chronic pain traits as estimated from the discovery cohort, we constructed four genetically independent pain phenotypes (GIPs; GIP1 to GIP4) in the discovery and replication cohorts. GIP1, explaining most of the genetic variance and covariance between the studied pain traits, was of foremost interest in the present research. Nevertheless, we also considered the remaining GIPs, which are genetically independent contributors to chronic pain at the four studied sites.

For each GIP, GWAS results were obtained. Associations reaching the genome-wide significance threshold in the discovery cohort were considered replicated if the Bonferroni-corrected significance threshold was reached in the meta-analysis of replication cohorts. For replicated loci, gene prioritization was performed using several approaches. We conducted a functional bioinformatics analysis searching for relevant gene sets and tissues (DEPICT/FUMA analyses), analyzed pleiotropic effects (SMR/HEIDI analysis) and investigated genetic correlations with other complex traits. In silico functional analysis was performed using the cohort of European ancestry individuals since this subsample was the largest.

**Genetically independent phenotypes**. The four original chronic musculoskeletal pain phenotypes were converted into GIPs using the coefficients of orthogonal transformation generated in the principal component analysis based on the matrix of genetic covariances. Coefficients of orthogonal transformation represent

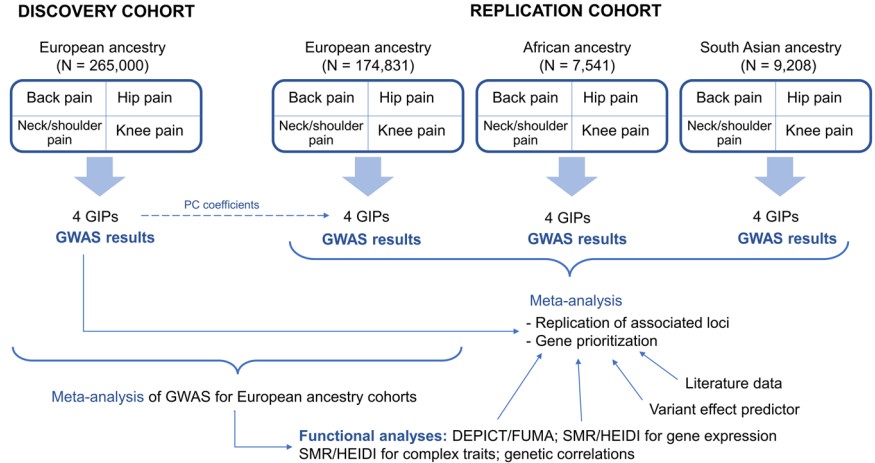

**Fig. 1 Overview of the study.** European ancestry individuals provided the matrix of genetic covariances and orthogonal transformation coefficients. The four chronic musculoskeletal pain phenotypes were decomposed into four GIPs. Orthogonal transformation coefficients were further used to construct GIPs in the replication cohorts of European, African, and South Asian ancestry individuals. For each GIP, GWAS results were obtained. Replication of associations and in silico functional analyses were based on the meta-analyses of GWAS for the replication cohorts and European ancestry cohorts, respectively. For replicated loci, the most likely causal genes were prioritized. DEPICT Data-driven Expression Prioritized Integration for Complex Traits framework, GIP genetically independent phenotype, PC principal components, SMR/HEIDI Summary data-based Mendelian Randomization analysis followed by the Heterogeneity in Dependent Instruments test, FUMA Functional Mapping and Annotation of Genome-Wide Association Studies platform.

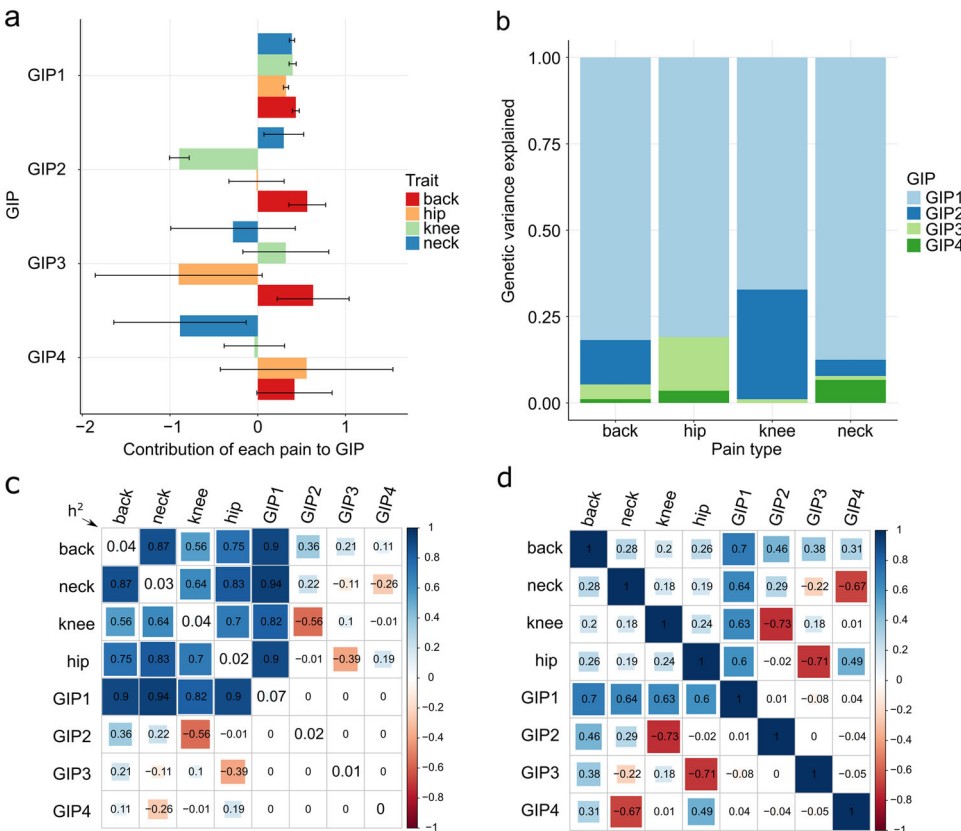

**Fig. 2 Genetically independent phenotypes (GIP) for chronic musculoskeletal pain. a** Barplots depicting the contribution of the four chronic musculoskeletal pain traits to each GIP. The bars represent orthogonal transformation coefficients, and the whiskers indicate their 95% confidence intervals. The violin plots depicting the empirical distribution of the coefficients of orthogonal transformation are presented in Supplementary Fig. 1. **b** Genetic variance of the studied chronic musculoskeletal pain explained by four GIPs. **c**. Estimated matrix of genetic correlations between the four chronic musculoskeletal pain phenotypes and GIPs. The diagonal elements represent LD Score regression estimates of SNP-based heritability ($h^2$) on the observed scale for each trait. **d** Matrix of phenotypic correlations between the four chronic musculoskeletal pain phenotypes and GIPs (estimated for pain phenotypes and predicted for GIPs, details are given in Supplementary Methods). Estimates for **c**, **d** were obtained using the discovery cohort of European ancestry individuals ($N = 265{,}000$).

contribution of each pain phenotype on each GIP, while genetic variance explained by GIPs approximates contribution of each GIP to each pain phenotype. A graphical representation of orthogonal transformation coefficients, as well as the genetic variance of chronic musculoskeletal pain phenotypes explained by each GIP, is shown in Fig. 2a, b, respectively. The violin plots of the empirical distribution of the coefficients of orthogonal transformation can be found in Supplementary Fig. 1.

The contributions of all pain phenotypes to GIP1 had the same direction and approximately the same magnitude. GIP1 showed the best stability based on the narrow 95% confidence intervals of orthogonal transformation coefficients. As expected, GIP1 explained the largest proportion of genetic variance (78.4%) of the four investigated musculoskeletal pain traits (the formula for calculating this value is provided in Supplementary Methods, page 9). LD Score regression-estimated SNP-based heritability of GIP1 was 7% on the observed scale and 15% on the liability scale and was found to be substantially larger than the heritability of the four individual pain phenotypes (2–4% on the observed scale and 7–9% on the liability scale, LD Score regression estimates, Fig. 2c, Supplementary Data 1a).

**GWAS for genetically independent phenotypes**. At the discovery stage, 9 loci passed the study-level threshold of statistical significance set at $P < 1.3e{-}08$ (5e-08/4, where 4 is the number of GIPs) after correction for the LD Score regression intercept (1.016 for GIP1, 1.001 for GIP2, 1.013 for GIP3, and 1.021 for GIP4). Six of the loci were associated with GIP1, and three with GIP2 (Table 1). Conditional and joint analysis showed single association signals per locus (Supplementary Data 2). A Manhattan plot of $-\log_{10}(P)$ for GIP1 is given in Fig. 3, Manhattan plots of $-\log_{10}(P)$ for GIP2–4 are given in Supplementary Fig. 2, quantile–quantile plots are presented in Supplementary Fig. 3, and regional association plots are shown in Supplementary Fig. 4.

Associations of six loci (five associated with GIP1 and one with GIP2) were replicated at $P < 5.6e{-}03$ (0.05/9, where 9 is the number of loci identified in the discovery stage) (Table 1). Full results of association analysis for each GIP and studied chronic musculoskeletal pain phenotype are provided in Supplementary Data 3.

Two of the six replicated loci showed genome-wide significant associations with chronic pain at specific location in the discovery cohort ($P < 5e{-}08$, Supplementary Data 3). These included the GIP1-associated locus near the *EXD3* gene (the locus is tagged by rs73581580 and is associated with chronic back pain with $P = 8.3e{-}09$) and the GIP2-associated locus near the *GDF5* gene (the locus is tagged by rs143384 and is associated with chronic knee pain with $P = 6.8e{-}16$ in our study and with knee pain in previous study[29]). In the meta-analysis of European ancestry discovery and replication cohorts, two additional loci reached a genome-wide significance for association with pain at specific location: the GIP1-associated locus near the *SLC39A8* gene (the locus is tagged by rs13107325 and is associated with chronic neck/shoulder pain with $P = 2.0e{-}08$) and the GIP1-associated locus near the *ECM1* gene (the locus is tagged by rs3737240 and is associated with chronic hip pain with $P = 8.4e{-}10$).

SNPs rs13107325, rs3737240, and rs143384 are known to have functional effects and/or to be associated with different complex traits and diseases. Summary data from published literature are provided in Supplementary Data 4.

**Functional annotation of the revealed signals**. *Literature-based gene prioritization*: for genes located near the lead SNPs (±250 kb) associated with GIPs, we performed a search in the Online Mendelian Inheritance in Man database (OMIM,

**Table 1 Top SNPs associated with GIPs.**

| GIP[a] | Lead SNP | Chr:position[b] | RefA/EffA[c] | Nearest gene[d] | Discovery cohort (N = 265,000) | | | | | Meta-analysis of 3 replication cohorts[f] | | | | |
|---|---|---|---|---|---|---|---|---|---|---|---|---|---|---|
| | | | | | β | SE | P | P[e](gc) | EAF (%) | β | SE | P | EAF (%) | N |
| **GIP2** | **rs143384** | **20:34025756** | **C/T** | **GDF5** | **−0.020** | **0.003** | **4.87e-13** | **7.40e-13** | **59.8** | **−0.022** | **0.003** | **1.65e-10** | **58.5** | **191,580** |
| **GIP1** | **rs7628207** | **3:49754970** | **T/C** | **AMIGO3** | **−0.023** | **0.004** | **1.71e-10** | **2.37e-10** | **82.3** | **−0.012** | **0.004** | **4.92e-03** | **81.8** | **191,580** |
| **GIP1** | **rs13107325** | **4:103188709** | **T/C** | **SLC39A8** | **−0.032** | **0.005** | **8.78e-10** | **1.19e-09** | **92.6** | **−0.035** | **0.007** | **4.21e-08** | **92.6** | **191,580** |
| **GIP1** | **rs3737240** | **1:150483355** | **T/C** | **ECM1** | **0.017** | **0.003** | **2.01e-09** | **2.69e-09** | **60.4** | **0.010** | **0.003** | **3.17e-03** | **61.1** | **191,580** |
| **GIP1** | **rs73581580** | **9:140251458** | **G/A** | **EXD3** | **0.025** | **0.004** | **3.89e-09** | **5.15e-09** | **12.4** | **0.030** | **0.005** | **9.54e-09** | **12.3** | **174,831** |
| **GIP1** | **rs12705966** | **7:114248851** | **G/A** | **FOXP2** | **0.018** | **0.003** | **5.71e-09** | **7.52e-09** | **66.7** | **0.012** | **0.004** | **1.70e-03** | **67.2** | **191,580** |
| GIP2 | rs4985445 | 16:69967835 | G/A | WWP2 | 0.017 | 0.003 | 1.56e-09 | 2.09e-09 | 54.3 | 0.007 | 0.003 | 0.0371 | 53.2 | 191,580 |
| GIP2 | rs548227718 | 5:175902724 | G/A | FAF2 | −0.283 | 0.048 | 3.02e-09 | 4.01e-09 | 0.1 | 0.096 | 0.060 | 0.1056 | 0.1 | 174,831 |
| GIP1 | rs111368900 | 1:53084695 | G/A | GPX7 | 0.242 | 0.041 | 5.01e-09 | 6.60e-09 | 0.2 | 0.089 | 0.048 | 6.55e-02 | 0.2 | 174,831 |

Replicated associations are shown in bold. EAF, effect allele frequency; SE, standard error; SNP, single nucleotide polymorphism.
EAF effect allele frequency, SE standard error, SNP single nucleotide polymorphism.
[a]Genetically independent phenotype with which the locus is associated.
[b]Chromosome: position on chromosome according to GRCh37.p13 assembly.
[c]Reference allele/effective allele.
[d]Nearest gene according to the NCBI dbSNP database (https://www.ncbi.nlm.nih.gov/snp/).
[e]P-value corrected for residual inflation using the LD Score regression intercept.
[f]Cohorts of individuals of African, South Asian, and European ancestry from the UK Biobank (3.9%, 4.8%, and 91.3% in the total replication cohort, N = 191,580).

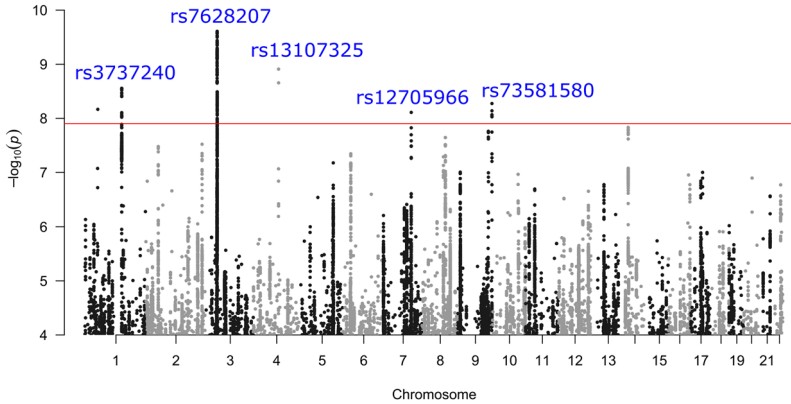

**Fig. 3 Graphical summary of the discovery GWAS results for GIP1 (European ancestry individuals, _N_ = 265,000).** Negative logarithms of *P*-values are presented after the genomic control correction using LD Score regression intercept. Only associations with *P* < 1.0e-04 are shown. Red line corresponds to the genome-wide significance threshold of *P* = 1.25e-08 (5.0e-08/4, where 4 is the number of GIPs). Replicated loci are annotated.

https://www.omim.org/), Google Scholar, the NCBI Gene (https://www.ncbi.nlm.nih.gov/gene), and the Pubmed database (https://www.ncbi.nlm.nih.gov/pubmed) to infer whether the biological functions of these genes may better explain their involvement in chronic musculoskeletal pain. The list of genes in the studied regions was based on regional association plots (Supplementary Fig. 4) and is given in Supplementary Data 3. Summary information on the genes that we considered most likely to be causal (literature data with references to corresponding sources) is provided in Supplementary Data 5.

*Prediction of SNP effects*: variant effect predictor (VEP) identified four missense variants: rs13107325 in the *SLC39A8* gene, rs3737240 and rs13294 in the *ECM1* gene, and rs79140116 in the *EXD3* gene. SIFT and PolyPhen tools predicted possibly damaging/deleterious effects only for rs13107325 and rs13294, while the remaining SNPs were designated as benign/tolerated (Supplementary Data 6a). Polymorphism rs13107325 is a triallelic SNP (C > T, A), and possibly damaging effects were predicted for both minor alleles T and A. Allele A is extremely rare and was not analyzed in the present study. Allele T was pain-predisposing (and positively associated with GIP1). Polymorphism rs13294 is also a triallelic SNP (G > A, T) and the extremely rare allele T was not covered by our GWAS. SIFT and PolyPhen tools predicted possibly damaging/deleterious effects only for the rare T variant, while allele A (inversely associated with GIP1 in our study) was attributed as benign/tolerated. However, it is still possible that in the case of a large effect of the rare allele rs13294T on GIP1, lead SNP rs3737240 only tags this rare variant (rs3737240 and rs13294 are located 1.6 kb from each other and are in high LD, $r^2 = 0.97$ in European ancestry populations). FATHMM-XF and FATHMM-INDEL identified a potentially pathogenic intronic SNP rs28535523 in the *UBA7* gene and an intronic indel rs34291892 in the *FOXP2* gene (Supplementary Data 6b, 6c). Potentially pathogenic variants rs28535523 T and rs34291892 insertion A were positively associated with GIP1. Data on matching the possibly damaging/deleterious/pathogenic alleles with the effects on GIPs, amino acid changes (where appropriate), and lead SNP alleles are presented in Supplementary Data 6d. All SNPs included in VEP and FATHMM analyses are listed in Supplementary Data 6e.

*Pleiotropic effects on gene expression*: the results of Summary data-based Mendelian Randomization (SMR) analysis followed by the Heterogeneity in Dependent Instruments (HEIDI) test are given in Supplementary Data 7 (associations that passed both SMR and HEIDI tests are presented in Supplementary Data 7a, full results are given in Supplementary Data 7b).

SMR/HEIDI analysis provided evidence that the same causal SNP in the locus tagged by rs143384 is associated with GIP2 and the expression of *GDF5*, *UQCC1*, and *RP3-477O4.16* (the gene encoding long intergenic non-coding RNA) in different tissues including brain caudate basal ganglia. Pleiotropic effects were also found for the locus tagged by rs3737240 associated with GIP1 and *MRPS21* gene expression in blood, and for the locus tagged by rs7628207 associated with GIP1 and expression levels of the genes *RBM6*, *FAM212A*, *RNF123*, and pseudogene *ACTBP13* (mainly in nervous tissues). It is likely that the locus tagged by rs7628207 contains regulatory elements that influence transcription of adjacent genes. Interestingly, *RNF123* gene expression has been linked to the risk of major depression[30], and major depressive disorders are genetically correlated with pain[25]. As the *AMIGO3* gene transcript (the CNS-related gene bearing the GIP1-associated SNP rs7628207 in its intron) was not present among the list of probes analyzed in the GTEx[31] and Westra projects[32], we could not infer pleiotropy. Other genes found in the literature-based and SNP effect analyses did not passed thresholds in SMR and HEIDI tests, signifying that we have no support to claim that their expression is influenced by causal variants associated with GIPs.

*DEPICT gene prioritization*: the results of DEPICT gene prioritization are given in Supplementary Data 8a (for input SNPs associated with GIPs at *P* < 1e-05) and Supplementary Data 8b (for input SNPs associated with GIPs at *P* < 5e-08). Statistically significant results (FDR < 0.05) were observed only for GIP1 and only when the *P*-value threshold for input SNPs was set at 1e-05. The list of prioritized genes is provided in Supplementary Data 8a. Of the genes identified in previous analyses, only *BSN* and *FOXP2* were found to be prioritized by DEPICT.

*Summary of gene prioritization*: a summary list of prioritized genes is presented in Table 2. For each locus tagged by rs143384, rs13107325, rs3737240, and rs12705966, two or more lines of evidence support a role for *GDF5*, *SLC39A8*, *ECM1*, and *FOXP2* genes, respectively, providing solid ground for their prioritization. Single candidate genes could not be suggested for loci tagged by rs7628207 and rs73581580 since different approaches yielded different results. The nearest gene to rs7628207 is *AMIGO3*, which has been shown to participate in inhibition of axon regeneration in the damaged CNS[33,34]. Five more genes are present in this region that were prioritized by in silico methods and/or based on prior literature data (in particular, the *BSN* gene encoding Bassoon presynaptic cytomatrix protein). Lead SNP rs73581580 is located in the intron of the *EXD3* gene, an ortholog

**Table 2 Summary of gene prioritization.**

| Lead SNP | Locus[a] | GIP[b] | Number of genes in the locus[c] | Prioritized gene | Nearest gene, yes/no (lead SNP location) | Evidence for prioritization |
|---|---|---|---|---|---|---|
| rs143384 | 20:34025756 | GIP2 | 15 | **GDF5** | Yes (5′ UTR) | L, S |
| rs7628207 | 3:49754970 | GIP1 | 18 | AMIGO3 | Yes (intronic) | L |
| | | | | BSN | No | L, D |
| | | | | RBM6 | No | S |
| | | | | FAM212A | No | S |
| | | | | RNF123 | No | S |
| | | | | UBA7 | No | V |
| rs13107325 | 4:103188709 | GIP1 | 3 | **SLC39A8** | Yes (missense) | L, V |
| rs3737240 | 1:150483355 | GIP1 | 19 | **ECM1** | Yes (missense) | L, V |
| rs73581580 | 9:140251458 | GIP1 | 32 | MIR7114 | No | L |
| | | | | NSMF | No | L |
| | | | | NOXA1 | No | L |
| | | | | GRIN1 | No | L |
| rs12705966 | 7:114248851 | GIP1 | 2 | **FOXP2** | Yes (intronic) | L, V, D |

Genes with strong evidence for prioritization are indicated in bold.
D DEPICT analysis, L literature-based prioritization (Supplementary Data 5), S SMR/HEIDI analysis, V variant effect predictor/FATHMM analysis, UTR untranslated region.
[a]Chromosome: position on chromosome according to GRCh37.p13 assembly.
[b]Genetically independent phenotype with which the locus is associated.
[c]Calculated based on regional association plots generated with LocusZoom tool (http://locuszoom.org/) in a 500-kb window (±250 kb around the lead SNP, Supplementary Fig. 4).

of *C. elegans* mut-7 gene required for transposon silencing and RNA interference in that organism. Nevertheless, results from other studies suggest four genes with more plausible effects on chronic musculoskeletal pain (*MIR7114*[35], *NOXA1*[36], *NSMF*[37,38], and *GRIN1*[39], Supplementary Data 5).

**Gene set and tissue/cell type enrichment.** DEPICT gene set and tissue/cell type enrichment analyses provided statistically significant results only for GIP1 (Supplementary Data 8c–f). For SNP sets associated with GIP1 with $P < 5e-08$, tissue/cell type enrichment with FDR < 0.05 was found for two terms: the "Neural Stem Cells" cell type and "Retina" tissue. However, relaxing the significance threshold of input SNPs to $P < 1e-05$ led to identification of 24 additional tissues, all of which were related to CNS. The same pattern was observed for gene set enrichment (for SNPs with $P < 1e-05$), revealing 462 terms mainly involved in nervous system function, development and morphology (e.g. "regulation of nervous system development", "axonogenesis", "synapse", and "regulation of transmission of nerve impulse").

FUMA gene set and tissue enrichment analyses for GIP1 detected 9 gene categories (6 of them were nervous system-related) and 12 brain tissues, respectively (Supplementary Data 9, Supplementary Fig. 5). For GIP2 and GIP3, a total of three gene sets were found by FUMA analysis, although we considered them as non-specific (e.g. "nikolsky_breast_cancer_20q11_amplicon"; Supplementary Data 9). No statistically significant gene sets were revealed for GIP4, and no statistically significant tissue types were identified for GIP2, GIP3, and GIP4.

**Pleiotropic effects on complex traits.** Five out of six replicated loci demonstrated pleiotropic effects on human complex traits in the SMR/HEIDI analysis (Supplementary Data 10, Fig. 4). As expected, the GIP1-associated locus rs13107325 (known as one of the most pleiotropic variants of the genome) was associated with the greatest number of diverse phenotypes, which included anthropometric traits (weight, height, and BMI), fluid intelligence score, prospective memory and education, sleep duration, Crohn's disease, self-reported osteoarthritis, diastolic blood pressure, blood cell traits, and alcohol intake frequency. Traits linked with the GIP2-associated locus rs143384 were mainly related to anthropometry and knee-related conditions (gonarthrosis and internal derangement of knee). The locus tagged by

the missense SNP rs3737240 (*ECM1* gene) showed pleiotropic effects on platelet count and plasma level of extracellular matrix protein 1 (ECM1) measured with the SOMAscan platform[40]. The same pain-promoting allele in this locus that was positively associated with GIP1 was linked to an increase in ECM1 level, reinforcing the role of *ECM1* as the candidate in this region. In the locus tagged by rs73581580, GIP1-associated alleles were linked to higher frequency of tiredness and difficulty of getting up in the morning. In the locus tagged by rs7628207, GIP1-associated variants were related to decreased plasma level of thioredoxin domain-containing protein 12 (TXNDC12), decreased overall health rating, decreased age at first live birth, decreased educational attainment, increased basal metabolic rate, and increased hip circumference. Interestingly, rs7628207 is adjacent to the *AMIGO3* gene prioritized by us based on the literature data (Table 2, Supplementary Data 5) which is linked to the gene encoding *TXNDC12* via a trans-protein QTL rs4688759[40].

Hospital-diagnosed osteoarthritis (the UK Biobank trait for which GWAS summary statistics were downloaded from the Michigan PheWeb database, see Methods section) was not revealed in the SMR/HEIDI analysis for any of the analyzed loci. However, for rs13107325, rs3737240, and rs143384, we can speculate that this could be due to the limited statistical power of the analysis. The SMR test $P$-values for these loci were quite low, although did not reach the Bonferroni-corrected significance threshold of $P = 3.71e-06$ (rs13107325: $P_{SMR} = 1.14e-05$, $beta_{SMR} = 0.63$; rs3737240: $P_{SMR} = 1.68e-05$, $beta_{SMR} = 0.89$; rs143384: $P_{SMR} = 6.13e-04$, $beta_{SMR} = -0.40$; $P_{HEIDI} \geq 0.01$ for all these loci). Thus, we cannot rule out a hypothesis that the same causal SNPs within the loci tagged by rs13107325 and rs3737240 may be associated with GIP1 and the increased risk of osteoarthritis, and the same causal SNPs within the locus tagged by rs143384 can be associated with GIP2 and the decreased risk of osteoarthritis.

**Genetic correlations between GIPs and complex traits.** GIP1 showed statistically significant genetic correlations with 40 complex traits (Fig. 5, Supplementary Data 11a; matrix of correlations between GIPs, chronic musculoskeletal pain phenotypes, and osteoarthritis is presented in Supplementary Fig. 6). Among them, 11 traits were directly linked to excess weight (BMI, overweight, obesity, and waist circumference), that is in line with known

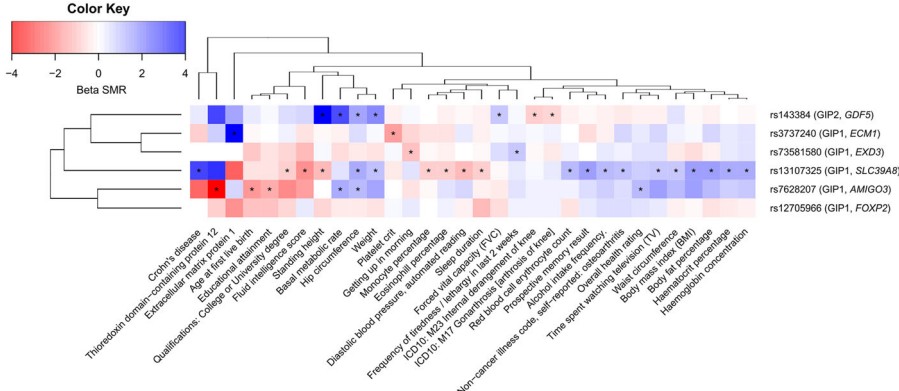

**Fig. 4 Pleiotropic effects of identified loci on human complex traits.** Color depicts the sign and the magnitude of SMR beta coefficient. Negative sign (red) means opposed effects on the corresponding GIP and the trait, and positive sign (blue) means the same direction of effect. |beta SMR| > 4 are depicted as |beta SMR| = 4. For "Prospective memory result" and "Overall health rating" trait, high scores correspond to poor performance. For "Getting up in morning" trait, high score corresponds to easy getting up. Traits that passed both SMR and HEIDI tests ($P_{SMR}$ < 3.71e-06 and $P_{HEIDI}$ ≥ 0.01) are marked with an asterisk. Data on 45 out of 78 revealed traits are not shown. Full results are given in Supplementary Data 10. GIPs associated with the loci and genes nearest to lead SNPs are indicated in parentheses. Dendrograms represent clustering based on complete linkage hierarchical clustering method.

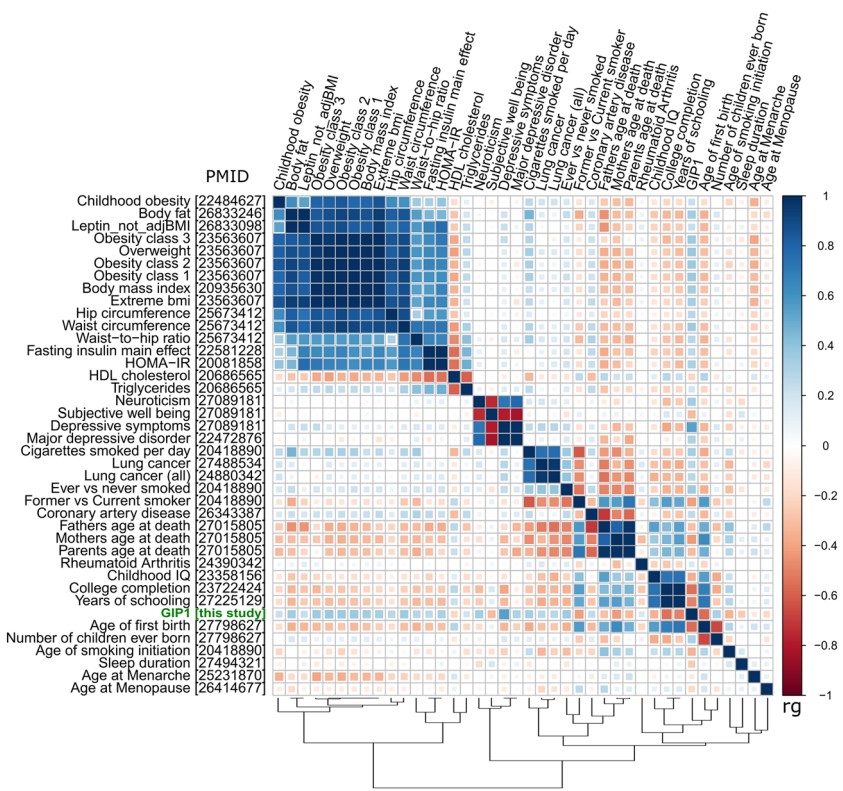

**Fig. 5 Matrix of genetic correlations between GIP1 and human complex traits.** Color depicts the sign and absolute value of the genetic correlation coefficients ($r_g$). Genetic correlations between GIP1 and all presented traits were statistically significant ($P$ < 5.98e-05). Osteoarthritis is not shown on this plot since genetic correlations analysis for this trait was performed using the GWAS-MAP platform, whereas for other traits, LD hub web interface was used. Matrix of genetic correlations between GIPs, chronic musculoskeletal pain traits and osteoarthritis is provided in Supplementary Fig. 6. HDL high density lipoprotein, HOMA-IR Homeostatic Model Assessment for Insulin Resistance, PMID PubMed ID number of the literature source providing GWAS summary statistics.

epidemiological associations between chronic pain and obesity-related traits[41]. Five more traits fell in the same cluster: HDL cholesterol (negative correlation with GIP1), triglycerides, HOMA-IR, leptin, and fasting insulin. Strong genetic correlations (|$r_g$| ranging between 0.31 and 0.54) were also revealed between

GIP1 and the cluster of psychiatric/personality traits (major depressive disorder, depressive symptoms, subjective well-being, and neuroticism). This finding is in accord with previous twin and family studies demonstrating a common genetic background for pain and depression[42–44]. Other traits included sociodemographic,

reproductive, education-related and smoking-related traits, osteoarthritis, rheumatoid arthritis, coronary artery disease, and sleep duration.

Traits that displayed the strongest genetic correlations with GIP1 were osteoarthritis (rg = 0.65), age of first birth (rg = −0.56), depressive symptoms (rg = 0.54), and college completion (rg = 0.54). Overall, the pattern of genetic correlations with GIP1 was very similar to that observed for back pain in our previous study[45]. GIP2 was genetically correlated only with osteoarthritis (inverse genetic correlation, rg = −0.30) and obesity-related traits, and GIP4 only with hip circumference (Supplementary Data 11b, c). No statistically significant genetic correlations with complex traits were found for GIP3 (Supplementary Data 11d).

Furthermore, we analyzed the genetic correlation between GIP1 and the first GIP constructed using the same methodology for a broader range of chronic pain traits (back, neck/shoulder, knee, hip, stomach/abdominal pain, and headache). We found out that these GIPs were almost genetically equivalent (rg = 0.99).

## Discussion

The genetic control of chronic musculoskeletal pain is complex, with each of very many genetic variants contributing a small effect. As a result, even very large genome-wide association studies provide only a limited number of replicated loci and rather low SNP-based heritability. Evidence from recent studies indicates that pain at different anatomical sites shares a common genetic component[24–26]. This suggests that combining several pain phenotypes in a single analytical framework may facilitate the discovery of common genetic factors – chronic musculoskeletal pain genes and pathways.

In the present study, we applied an approach that allowed us to detect genes shared between four common chronic musculoskeletal pains: back, neck/shoulder, knee, and hip. Our approach relies on capturing heredity of a set of genetically correlated traits via constructing genetically independent phenotypes (GIPs; Fig. 2a). The GIPs are defined as a weighted sum of the original phenotypes, with weights selected in such a way that the first GIP (GIP1) explains most genetic variance of and covariance between the studied traits, with the later GIPs (GIP2–4) explaining progressively less. The four weights defining GIP1 based on the four chronic pain traits (back, neck/shoulder, hip, and knee pain) turned out to be approximately the same (Fig. 2a, Supplementary Fig. 1). This means that GIP1, the genetic component explaining most of the cases of chronic musculoskeletal pain at the studied sites, affects the risk of chronic musculoskeletal pain to approximately the same degree, irrespective of pain's location. Unlike the first GIP, the second GIP is site-specific and reflects a genetic propensity for knee pain, but not the back or neck/shoulder pain.

We mapped and replicated six genomic loci (five associated with GIP1 and one with GIP2). Importantly, in the discovery sample, only two out of six replicated loci were genome-wide significantly associated with the individual pain phenotypes: rs73581580 with chronic back pain and rs143384 with chronic knee pain. Also, as expected, the SNP-based heritability of GIP1 was substantially higher than for any of separate pain traits (7% vs 2–4%, observed scale, LD Score regression estimates). These results highlight the improved power of the GIP approach for identifying genetic predictors of chronic pain predisposition. It should be noted that phenotypic correlations between the traits were much lower than the genetic correlations (Fig. 2c, d; pairwise phenotypic correlations ranged from 0.18 to 0.28, while pairwise genetic correlations ranged from 0.56 to 0.87). In this scenario, we can speculate that conventional multivariate approaches based on phenotypic correlations like MANOVA[46] or

MultiPhen[47] would have been less powerful than our method based on genetic correlations. Moreover, while estimation of phenotypic correlation is impossible for non-overlapping samples, genetic correlations can be calculated for both overlapping samples and independent cohorts[48]. This makes our approach applicable to the traits measured within the frameworks of different genomics consortia.

Among the six replicated loci, three were well-studied polymorphisms associated with different traits and conditions in previous works (rs13107325, rs3737240, and rs143384, Supplementary Data 4). In the present study, we performed a hypothesis-free analysis of pleiotropic effects of six GIP-associated loci on 2243 complex human traits. Our analysis revealed 78 phenotypes influenced by the same causal polymorphisms that are associated with GIPs (Supplementary Data 10, Fig. 4). These phenotypes included a broad variety of anthropometric, sociodemographic, behavior and personality traits, diseases (such as Crohn's disease, gonarthrosis and osteoarthritis), and laboratory parameters. Interestingly, GIP1-associated alleles in the locus tagged by rs73581580 were also associated with higher frequency of tiredness and difficulty of getting up in the morning. Our results demonstrate diversity of effects of the GIP-associated loci and suggest the presence of common pathways underlying chronic musculoskeletal pain and multiple other human traits.

GIP1-associated pathways and tissues were mostly related to CNS development and functioning, suggesting that GIP1 depicts neurological and psychological components of chronic pain. Consistent with this, one of the genes prioritized for GIP1-associated loci based on multiple lines of evidence was FOXP2, whose product is a transcription factor expressed in fetal and adult brain and required for the development of speech and language regions[49,50]. Involvement of psychological component in chronic pain was additionally supported by the finding of a very strong positive genetic correlation between GIP1 and depressive symptoms. Having said that, it is equally important that GIP1 was associated also with traits reflecting general health and risk factors for musculoskeletal pain: sociodemographic, reproductive, education- and smoking-related traits, and sleep duration. Importance of morphological factors for chronic musculoskeletal pain was also demonstrated by revealing of GIP1-associated genes SLC39A8 and ECM1, which are known to be implicated in the development and functioning of the musculoskeletal system. ECM1 gene encodes a negative regulator of bone mineralization and chondrogenesis[51–53]. GIP1-associated ("pain-promoting") variant in this gene showed an association with the increased level of ECM1 protein in our SMR/HEIDI analysis. GIP1-associated ECM1 allele rs3737240 C is in a high LD ($r^2 = 0.94$ in European ancestry populations) with the allele rs12040949 C, which was associated with the increased risk of hip osteoarthritis in a recent study[54]. The product of the SLC39A8 gene was shown to participate in osteoarthritis cartilage destruction[55,56]. Slc39a8 mutant zebrafish exhibit vertebral abnormalities, impaired growth, and decreased motor activity, and a missense GIP1-associated polymorphism rs13107325 in the SLC39A8 gene has previously been associated with the increased risk of osteoarthritis[54] and severe adolescent idiopathic scoliosis[57]. Thus, similar to findings from our recent study of back pain[45], genetic factors underlying chronic musculoskeletal pain comprise biological, social, and psychological components.

Since our study was aimed at investigating chronic musculoskeletal pains at anatomical sites commonly affected by osteoarthritis, it was not surprising that we found loci and genes associated with this condition and found high genetic correlation between osteoarthritis and GIP1 (rg = 0.65). Note, that this genetic correlation is similar in magnitude to correlation between

GIP1 and age of first birth (−0.56), indicating that although similarities are high, there exist substantial differences between OA and GIP1. Furthermore, for GIP1, gene/tissue enrichment analysis revealed a plethora of CNS-related terms. In a recent large-scale genetic study for OA, enriched terms were not directly linked to the nervous system ("anatomical structure morphogenesis", "ion channel transport", "histidine metabolism", etc.)[54]. Finally, we performed genetically independent phenotype analysis for the extended set of chronic traits, which include not only musculoskeletal pain (six traits: back, neck/shoulder, knee, hip pain as well as stomach/abdominal pain and headache). Genetic correlation between GIP1 for four pain traits and GIP1 for six pain traits was extremely high (rg = 0.99) providing evidence that, despite high genetic overlap with OA, GIP1 for musculoskeletal pain may reflect chronic pain per se.

It is noteworthy that pain is the main symptom and clinical outcome of osteoarthritis. In the UK Biobank study which provided GWAS summary statistics for OA[58], phenotypes were defined according to ICD-9/ICD-10 codes (electronic medical record data), so whether the study participants were examined radiographically or not is unknown. Thus, genetic overlap between GIP1 and OA can be actually biased by a genetic correlation between GIP1 and not the OA, but pain in OA. Besides this, a study by Valdes et al.[59] obtained interesting results on the inverse relationship between preoperative radiographic severity and postoperative pain in OA patients who have undergone total joint replacement (TJR). We hypothesized that in OA patients with low preoperative radiographic damage, pain leading to TJR can be caused not entirely by a joint damage, but also by other factors such as central sensitization. It is possible that these factors have common genetic background with GIP1 constructed in our study.

Given that GIP1 essentially contrasts chronic musculoskeletal pain (in general) with an unpainful state, the other GIPs might be expected to account for musculoskeletal pain at specific anatomical locations. This was indeed the case with GIP2, which had the greatest impact on knee pain (Fig. 2b). The only gene found to be associated with GIP2 at the genome-wide significance level was GDF5, a gene with well-established associations with peripheral osteoarthritis and intervertebral disc degeneration[54,60–65]. These results are consistent with the fact that the knee is one of the most common sites of osteoarthritis. For GIP3 and GIP4, no firm conclusions can be drawn regarding what component of pain they might represent, but, as can be seen from Fig. 2b, GIP3 makes a substantial contribution to hip pain and GIP4 to neck/shoulder pain.

Another approach recently applied in GWAS of chronic pain is based on obtaining a phenotype of multisite chronic pain (MCP) as a sum of the number of anatomical sites affected by pain (a study by Johnston et al.[66]). Johnston et al. carried out a large-scale GWAS of MCP in ~380,000 UK Biobank participants. They obtained results supporting the hypothesis that chronic pain involves a strong nervous system component and demonstrated a causal effect of MCP on major depressive disorder. However, the summing of different pain sites into a quantitative MCP phenotype assumes equivalence between the genetic predictors of musculoskeletal pain conditions (such as back and knee pain) and the genetic predictors of non-musculoskeletal pain conditions that may include substantial components of pain due to other causes, such as migraine (in the case of headache), dental or neuropathic pain (in the case of facial pain), or visceral pain (in the case of stomach/abdominal pain). Such equivalence may be too strong an assumption to make without empirical justification. Our approach is empirical, with definition of GIPs driven by the data; another strength of our approach is its ability to reveal pain type specific genetic loci as exemplified by GDP5 associated with

GIP2 representing knee pain. Comparing with a direct knee pain GWAS, GIP2 may provide a more knee-specific phenotype from which general propensity to pain is subtracted. This claim requires experimental validation, though.

Nevertheless, as far as we are aware, our study, together with that by Johnston et al.[66], is among the first to use a GWAS framework to address the genetics of chronic pain at multiple sites. Despite the difference in methodology and the phenotypes involved, our study identified five loci also reported by Johnston et al.: AMIGO3 (tagged by rs7628207 in Johnston et al.[66]), SLC39A8 (tagged by rs13135092 in Johnston et al.[66]), ECM1 (tagged by rs59898460 in Johnston et al.[66]), EXD3 (tagged by rs73581580 in both Johnston et al.[66] and our study), and FOXP2 (tagged by rs12537376 in Johnston et al.[66]). It should be noted that in our study, in contrast to the study by Johnston et al., these loci have been replicated. However, both discovery and replication stages in our study as well as analyses conducted by Johnston et al. were based on the UK Biobank data only, highlighting the need to replicate these findings in independent cohorts.

Our study has limitations. The first general limitation is related to a questionnaire-based approach to phenotyping, which may lead to heterogeneous pain phenotypes. Our methods attempted to overcome this by constructing genetically independent phenotypes whose genetic basis approximates the genetic background of distinct phenotypes and likely represents the "general pain" component of analyzed musculoskeletal pain traits. Second, in our study, we focused only on chronic musculoskeletal pain at anatomical sites potentially linked through osteoarthritis, so one must be cautious generalizing our results to other chronic pain conditions. Third, even though we carried out replication analysis, the replication cohorts were drawn from the same source dataset (UK Biobank), so sampling bias cannot be excluded. Finally, for two out of six identified loci (tagged by rs7628207 and rs73581580), we were not able to prioritize a single causal gene, and candidate genes suggested for the locus tagged by rs73581580 were selected based only on data from available literature sources.

In summary, our study of genetically independent components of chronic musculoskeletal pain phenotypes revealed hereditary factors shared by chronic back, neck/shoulder, hip, and knee pain and identified loci and genes relevant for these conditions. Our results provided further support that neurological and psychological components are important contributors to chronic pain. Using this approach may facilitate discovery of chronic pain mechanisms.

## Methods

**Study sample and phenotype definition.** The study sample comprised UK Biobank participants[28]. Sociodemographic, physical, lifestyle, and health-related characteristics of this cohort have been reported elsewhere[67]. In brief, individuals enrolled in the UK Biobank study were aged 40–69 years; were less likely to be obese, to smoke, to drink alcohol; had fewer self-reported health conditions as compared to the general population. All study participants provided written informed consent, and the study was approved by the North West Multi-Centre for Research Ethics Committee (11/NW/0382).

This particular study was approved by the UK Biobank research team under project #18219. Cases and controls were defined based on questionnaire responses. First, participants responded to "Pain type(s) experienced in the last months" followed by questions inquiring if the specific pain had been present for more than 3 months. Those who reported back, neck or shoulder, hip, or knee pain lasting more than 3 months were considered chronic back, neck/shoulder, hip, and knee pain cases, respectively. Participants reporting no such pain lasting longer than 3 months were considered controls (regardless of whether they had another regional chronic pain, such as abdominal pain, or not). Individuals who preferred not to answer were excluded from the study. Besides this, we excluded individuals who reported more than 3 months of pain all over the body because this phenotype was by definition negatively correlated with other pain phenotypes (those who have answered "pain all over the body" could not specify any location of pain) and had a low prevalence in the UK Biobank cohort (1.6%). Further details of phenotype definitions are given in Supplementary Methods.

**Table 3 Descriptive characteristics of the study cohorts.**

|  | Prevalence | Sample size | Age (mean ± SD) (years) | BMI (mean ± SD) (kg m$^{-2}$) | Women (%) |
|---|---|---|---|---|---|
| **Discovery cohort[a] (N = 265,000)** |  |  |  |  |  |
| Chronic back pain | 17.9% | Cases (N = 47,507) | 57.65 (7.99) | 28.33 (5.18) | 53.88 |
|  |  | Controls (N = 217,493) | 57.26 (8.03) | 27.15 (4.61) | 54.32 |
| Chronic neck pain | 16.3% | Cases (N = 43,287) | 57.73 (7.79) | 27.90 (5.02) | 53.84 |
|  |  | Controls (N = 221,713) | 57.25 (8.07) | 27.25 (4.68) | 54.32 |
| Chronic hip pain | 9.2% | Cases (N = 24,300) | 59.15 (7.44) | 28.91 (5.40) | 54.35 |
|  |  | Controls (N = 240,700) | 57.15 (8.06) | 27.20 (4.64) | 54.23 |
| Chronic knee pain | 17.5% | Cases (N = 46,292) | 58.61 (7.59) | 29.18 (5.37) | 54.12 |
|  |  | Controls (N = 218,708) | 57.06 (8.09) | 26.97 (4.50) | 54.27 |
| **Replication cohort (N = 191,580)** |  |  |  |  |  |
| **African ancestry (N = 7541)** |  |  |  |  |  |
| Chronic back pain | 21.0% | Cases (N = 1586) | 53.77 (8.24) | 30.62 (5.79) | 54.50 |
|  |  | Controls (N = 5955) | 52.04 (8.00) | 29.27 (5.13) | 54.19 |
| Chronic neck pain | 16.1% | Cases (N = 1217) | 54.38 (7.98) | 30.06 (5.52) | 54.35 |
|  |  | Controls (N = 6324) | 52.02 (8.04) | 29.45 (5.25) | 54.24 |
| Chronic hip pain | 8.5% | Cases (N = 641) | 55.00 (7.91) | 31.30 (6.14) | 54.37 |
|  |  | Controls (N = 6900) | 52.16 (8.05) | 29.39 (5.19) | 54.25 |
| Chronic knee pain | 20.4% | Cases (N = 1539) | 54.67 (8.30) | 31.64 (6.11) | 54.49 |
|  |  | Controls (N = 6002) | 51.82 (7.92) | 29.01 (4.93) | 54.20 |
| **European ancestry (N = 174,831)** |  |  |  |  |  |
| Chronic back pain | 18.0% | Cases (N = 31,428) | 57.62 (7.96) | 28.36 (5.22) | 54.05 |
|  |  | Controls (N = 143,403) | 57.26 (8.02) | 27.14 (4.58) | 54.28 |
| Chronic neck pain | 16.3% | Cases (N = 28,482) | 57.82 (7.76) | 27.92 (5.02) | 54.27 |
|  |  | Controls (N = 146,349) | 57.23 (8.06) | 27.25 (4.66) | 54.24 |
| Chronic hip pain | 9.2% | Cases (N = 16,022) | 59.26 (7.40) | 28.86 (5.41) | 54.61 |
|  |  | Controls (N = 158,809) | 57.13 (8.05) | 27.21 (4.63) | 54.20 |
| Chronic knee pain | 17.3% | Cases (N = 30,173) | 58.71 (7.54) | 29.24 (5.41) | 54.27 |
|  |  | Controls (N = 144,658) | 57.04 (8.08) | 26.97 (4.47) | 54.23 |
| **South Asian ancestry[b] (N = 9208)** |  |  |  |  |  |
| Chronic back pain | 21.6% | Cases (N = 1993) | 54.66 (8.51) | 27.76 (4.58) | 54.29 |
|  |  | Controls (N = 7215) | 53.87 (8.47) | 26.92 (4.23) | 54.22 |
| Chronic neck pain | 20.2% | Cases (N = 1864) | 54.65 (8.24) | 27.43 (4.56) | 54.31 |
|  |  | Controls (N = 7344) | 53.88 (8.53) | 27.01 (4.25) | 54.22 |
| Chronic hip pain | 6.6% | Cases (N = 610) | 56.61 (8.21) | 28.30 (4.90) | 54.07 |
|  |  | Controls (N = 8598) | 53.86 (8.47) | 27.01 (4.26) | 54.25 |
| Chronic knee pain | 20.1% | Cases (N = 1850) | 55.97 (8.23) | 28.52 (4.86) | 54.10 |
|  |  | Controls (N = 7358) | 53.55 (8.47) | 26.74 (4.09) | 54.27 |

[a]Discovery cohort comprised only individuals of European ancestry.
[b]Indian, Pakistani, and Bangladeshi.

Overall, 456,580 individuals with imputed genotype data and phenotype data were included in the present study. Of these, 265,000 participants of European ancestry (defined by SNP-based principal component analysis) were randomly selected to provide the GWAS discovery cohort. The decision to include only Europeans was based solely on the highest representation of these individuals among the UK Biobank participants. The replication cohort (N = 191,580) comprised individuals of African (N = 7,541) and South Asian ancestry (Indian, Pakistani, and Bangladeshi; N = 9,208) as well as the remaining European ancestry participants (N = 174,831). Descriptive characteristics of the groups is provided in Table 3. Splitting the European ancestry sample by 265,000 and 174,834 was done in order to obtain the optimal balance of statistical power between discovery and replication stages. We estimated the statistical power for different sample sizes of discovery and replication cohorts (based on the effect sizes of the top SNPs from our previous back pain GWAS[45]) and showed that splitting by 3/5 for discovery and 2/5 for replication would be an optimal ratio since it enabled reaching 80% power at both stages.

**Genotyping and imputation**. Genotyping and imputation data were obtained from the UK Biobank March 2018 data release. Genotyping was conducted using the Affymetrix UK BiLEVE and Affymetrix UK Biobank Axiom arrays. Imputation was performed with the IMPUTE4 program (https://jmarchini.org/impute-4/)[68] using the Haplotype Reference Consortium (HRC)[69] and merged UK10K and 1000 Genomes phase 3 reference panels. Details on DNA extraction and quantification[70] as well as on the centralized analysis of the genetic data, genotype quality, properties of population structure and relatedness of the genetic data, and efficient phasing and genotype imputation have been reported previously[68].

**Genome-wide association study**. GWAS were carried out using BOLT-LMM v.2.3.2 software[71]. Linear mixed-effects models were fitted to test for additive effects of the SNPs (genotype dosage) on pain phenotypes adjusting for age, sex, genotyping platform batch, and the first ten genetic principal components. The following filters were applied: minor allele frequency >0.0002 for Europeans and >0.005 for Africans and Asians; imputation quality score >0.7; genotyping and individual call rates >0.98. Only biallelic autosomal SNPs and indels were analyzed. BOLT-LMM software requires LD score data for the analysis. For Europeans, we used LD scores distributed as part of BOLT-LMM package. For Africans and South Asians, we carried out LD score estimation using LD score software[72] and data from 500 individuals randomly selected from each ethnic group. The results of GWAS were corrected for residual inflation using the LD score regression intercept[72].

**Locus definition**. Associated loci were defined as regions within ±250 kb around the lead SNP. Only the most significant SNP per locus was reported.

**Genetically independent phenotypes**. To elucidate genetic components explaining four chronic musculoskeletal pain phenotypes (chronic back, neck/shoulder, hip, and knee pain), we used a modified principal component analysis (PCA) technique that combines multiple correlated variables into a set of uncorrelated principal components (PCs). PCs are linear combinations of variables constructed such that the first PC explains the maximum proportion of the total variance of the set of traits, the second PC accounts for the largest proportion of the remaining variance, and so on. In conventional PCA of a set of traits, vectors of coefficients of orthogonal transformation are equal to the eigenvectors of the matrix of phenotypic covariance. In the present study, we used the matrix of

genetic covariances between the traits of interest to decompose them into genetically independent components, that we called genetically independent phenotypes (GIPs). GIPs are not correlated genetically and the first GIP (GIP1) explains most of the genetic variance of -and covariance between- four musculoskeletal pain phenotypes. Technical details of our approach are described in Supplementary Methods. The method can be applied for three or more genetically correlated traits. The GIP1 captures large part of the variation for each trait in case of high genetic correlations between traits (see examples in Supplementary Methods). It should be noted that principal component analysis has already been used for studying genetic background of complex traits[73,74], although it was applied to obtain phenotypically independent phenotypes, not GIPs. In both cases heritability of obtained principal components was not less than heritability of original traits.

The matrix of genetic covariances (estimated by LD Score regression[48]) and orthogonal transformation coefficients were obtained using the discovery cohort of European ancestry individuals. The 95% confidence intervals of these coefficients were estimated via the Monte Carlo sampling. For each resulting "discovery" GIP, GWAS results were calculated as described in Supplementary Methods.

GIPs for replication datasets were constructed using the orthogonal transformation coefficients obtained at the discovery step. GWAS results for each "replication" GIP were combined by a meta-analysis. Furthermore, GWAS for GIPs for European ancestry replication cohort ($N = 439,831$ in total) were meta-analyzed with GWAS for discovery GIPs, and the results were used for subsequent post-GWAS in silico analyses. Meta-analyses were conducted using the inverse-variance-weighted approach (fixed-effects model) with METAL software[75].

Additionally, we used the same methodology to obtain the first GIP for the extended set of pain traits available in the UK Biobank: chronic back, neck/shoulder, hip, knee, stomach/abdominal pain and headache. Facial pain, which is also present in the UK Biobank database, was not included in the analysis due to low prevalence (0.9% in European ancestry dataset, 4016 cases and 435815 controls) and statistically insignificant SNP-based heritability, that makes the genetic correlation analysis impossible. GIP1 for six pain phenotypes was constructed for the discovery and European ancestry replication cohort, and GWAS results for these cohorts were meta-analyzed. GIP1 for six pain phenotypes was included in the analysis of genetic correlation with GIP1 for four pain phenotypes.

**Conditional analysis.** Conditional and joint (COJO) analysis was carried out as previously described[76]. Calculations were performed using the GCTA software[77]. Linkage disequilibrium (LD) matrix was computed with PLINK 1.9 software (https://www.cog-genomics.org/plink2) using 100,000 individuals randomly selected from the discovery cohort. We claimed one independent signal per locus if no polymorphism other than the lead SNP passed the significance threshold of $P = 5e-08$. Regional association plots were generated using LocusZoom (http://locuszoom.org/) for regions within ±250 kb from the lead SNP.

**Prediction of SNP effects.** We analyzed the functional effects of a set of SNPs and indels in high LD ($r^2 > 0.8$) with replicated variants. LD was calculated using PLINK 1.9[78] (—show-tags option) and genotype data for 503 European ancestry individuals (1000 Genomes phase 3 version 5 data). Additionally, we selected SNPs within replicated regions (±250 kb from lead SNPs) associated with GIPs at $P \leq T$, where $\log10(T) = \log10(P_{min}) + 1$, and $P_{min}$ is a $P$-value for the strongest association per locus. These SNPs were added in the analysis since genotype data for the UK Biobank samples were imputed using the Haplotype Reference Consortium (HRC) panel, and some HRC SNPs could possibly be missed in the 1000 Genomes panel. All selected variants were annotated using the Ensembl Variant Effect Predictor (VEP)[79] as well as FATHMM-XF[80] and FATHMM-INDEL[81]. In the latter two methods, predictions of variant effects were made according to scores ranging from 0 to 1, with scores above 0.5 predicted to be deleterious while those below 0.5 predicted to be neutral or benign.

**DEPICT and FUMA analyses.** Gene set and tissue/cell type enrichment analyses and gene prioritization were performed using the Data-driven Expression Prioritized Integration for Complex Traits (DEPICT) tool[82]. We employed the DEPICT software version 1.1, release 194 with default parameters (https://data.broadinstitute.org/mpg/depict/). Tests were conducted for both genome-wide significant SNPs ($P < 5e-08$) and for SNPs associated with GIPs at $P < 1e-05$. The MHC region was omitted. The significance threshold for DEPICT analyses was set at FDR < 0.05.

Gene set and tissue enrichment analyses were also performed using the FUMA (Functional Mapping and Annotation of Genome-Wide Association Studies) platform[83] (GENE2FUNC function, with default parameters) based on the MAGMA method[84] and the MsigDB c5 database[85]. The significance threshold for FUMA analyses was set at Bonferroni-corrected $P$-value < 0.05.

**SMR/HEIDI analysis.** Summary data-based Mendelian Randomization (SMR) analysis followed by the Heterogeneity in Dependent Instruments (HEIDI) test[86] was used to study potential pleiotropic effects of identified loci on GIPs, human complex traits, and gene expression levels in different tissues. SMR analysis provides evidence for pleiotropy (the same locus is associated with two or more traits).

It cannot define whether traits in a pair are affected by the same underlying causal polymorphism, and this is specified by a HEIDI test, which distinguishes pleiotropy from linkage disequilibrium. It should be noted that SMR/HEIDI analysis does not identify which allele is causal and cannot distinguish pleiotropy from causation.

Summary statistics for gene expression levels were obtained from Westra Blood eQTL (peripheral blood, http://cnsgenomics.com/software/smr/#eQTLsummarydata)[32] and the GTEx version 7 database (48 tissues, https://gtexportal.org)[31]. Summary statistics for other complex traits were derived from the GWAS-MAP database[87] developed by our group. The GWAS-MAP platform integrates a database of summary-level GWAS results for 673 complex traits from the UK Biobank, 123 metabolomics traits, 1206 circulating proteins, 41 cytokines and growth factors, 190 plasma protein and IgG N-glycosylation traits, inflammatory bowel disease (including Crohn's disease), and 8 traits related to coronary artery disease, myocardial infarction, and factors associated with these conditions. Summary statistics for the UK Biobank traits were provided by the Neale Lab (http://www.nealelab.is/) and the Gene ATLAS (http://geneatlas.roslin.ed.ac.uk/)[88]. In this study, we added to the GWAS-MAP database results from 18 GWAS of chronic musculoskeletal pain-related traits obtained in the present study (GWAS in the discovery dataset and the results from European ancestry meta-analysis for chronic back, neck/shoulder, knee, hip pain; GWAS in the discovery dataset and the results from European ancestry meta-analysis for GIPs constructed for these four phenotypes; European ancestry meta-analysis of GWAS for chronic stomach/abdominal pain and chronic headache. Additionally, we added the results of GWAS of osteoarthritis from the Michigan PheWeb database (http://pheweb.sph.umich.edu/SAIGE-UKB/pheno/740). This OA GWAS was performed using the UK Biobank data by the Scalable and Accurate Implementation of GEneralized mixed model (SAIGE) method[58].

Description of all 2262 traits is provided in Supplementary Data 1b. The GWAS-MAP platform contains embedded software for our implementation of SMR/HEIDI analysis[86], LD Score regression[72], and 2-sample Mendelian randomization analysis (MR-Base package[89]). Further details are given in Supplementary Methods.

In gene expression analysis, the significance threshold for SMR was set at $P = 3.24e-06$ (0.05/15,445, where 15,445 is the total number of tests corresponding to all analyzed SNPs, expression probes, and tissues). In complex traits analysis, the significance threshold for SMR was set at $P = 3.71e-06$ (0.05/(6 × 2244), where 6 is the number of loci, and 2244 is the number of non-pain traits). The significance threshold for HEIDI tests in both analyses was set at $P = 0.01$ ($P < 0.01$ corresponds to the rejection of pleiotropy hypothesis). Details of data processing are given in Supplementary Methods.

**Genetic correlations and heritability.** SNP-captured heritability (h2) and genetic correlations between GIPs and human complex traits were estimated using LD Score regression[48]. SNP-captured heritability (h2) of pain traits was also estimated using the restricted multiple likelihood (REML) algorithm in BOLT-LMM v.2.3.2 software[71]. In total, we examined 209 non-UK Biobank traits available in the LD hub database (http://ldsc.broadinstitute.org/ldhub/). We removed duplicates and included only the most recent study for each trait (as indicated by the largest PubMed ID number). Since osteoarthritis was not present in the LD hub database, we used summary statistics for this trait obtained from the Michigan PheWeb database (http://pheweb.sph.umich.edu/SAIGE-UKB/pheno/740). The statistical significance threshold was set at 5.95e-05 (0.05/(210 × 4), where 210 is the number of traits and 4 is the number of GIPs).

Genetic correlations between GIPs and LD hub traits were calculated using the LD hub web interface. Genetic correlations between GIPs, osteoarthritis and chronic pain traits were calculated using the GWAS-MAP platform.

For 39 LD hub traits showing statistically significant correlations with GIP1 as well as for osteoarthritis, four chronic pain traits and four GIPs, matrices of genetic correlation were generated. Clustering and visualization were performed by the "corrplot" package for the R language (basic "hclust" function). For clustering, we estimated squared Euclidean distances by subtracting absolute values of genetic correlation from 1 and used the Ward's clustering method.

Additionally, we estimated the genetic correlation between GIP1 for four analyzed chronic pain traits and the first GIP constructed using the same methodology for six chronic pain traits (back, neck/shoulder, knee, hip, stomach/abdominal pain, and headache) using the GWAS-MAP platform.

**Reporting summary.** Further information on research design is available in the Nature Research Reporting Summary linked to this article.

## Data availability

UK Biobank data are available upon application. Summary statistics from the GWAS reported in this study are available for download from Zenodo[90] under the CC BY 4.0 license.

## Code availability

All computer code used in this research is available at https://github.com/Sodbo/Pain3_project_code[91].

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

## Acknowledgements

The work of Y.S.A. and S.Z.S. was supported by the Russian Ministry of Education and Science under the 5–100 Excellence Programme and by the Federal Agency of Scientific Organizations via the Institute of Cytology and Genetics (project 0324-2019-0040-C-01 / AAAA-A17-117092070032-4). The work of Y.A.T., A.S.S., and E.E.E. was supported by the Russian Foundation for Basic Research (project 19-015-00151). The contribution of L.C.K. was funded by PolyOmica. P.S. was supported by VA Career Development Award # 1IK2RX001515 from the United States (U.S.) Department of Veterans Affairs Rehabilitation Research and Development (RR&D) Service. P.S. is a Staff Physician at the VA Puget Sound Health Care System. The contents of this work do not represent the views of the U.S. Department of Veterans Affairs or the United States Government. This study was approved by the UK Biobank research team under project #18219.

## Author contributions

Y.S.A., F.M.K.W., and P.S. conceived and oversaw the study. Y.A.T., M.B.F., A.S.S., S.Z.S., E.E.E., J.Z., Y.S.A., F.M.K.W., and P.S. contributed to the design of the study and interpretation of the results. Y.A.T., M.B.F., S.Z.S., and E.E.E. carried out statistical analysis. A.S.S. performed literature analysis and literature-based gene prioritization and produced the first draft of the manuscript. L.C.K. provided statistical and computational support. All co-authors discussed the results and contributed to preparing the final version of the manuscript.

## Competing interests

Y.S.A. and L.C.K. declare the following competing interests: Y.S.A. and L.C.K. are owners of Maatschap PolyOmica and PolyKnomics B.V., private organizations, providing services, research and development in the field of computational and statistical, and quantitative and computational (gen)omics. Y.A.T., M.B.F., A.S.S., S.Z.S., E.E.E., J.Z., P.S., and F.M.K.W. declare no competing interests.
