## [Peer Review File · Communications Biology]

Reviewers' comments:

Reviewer #1 (Remarks to the Author):

Tsepilov et al has prepared a manuscript that aims to identify shared genetic components among chronic musculoskeletal pain occurred at different body sites. They adapted a new concept of genetically independently phenotypes (GIPs) to replace the traditional 'Yes or No' pain phenotypes at 4 body sites. It is a new attempt and should be encouraged since recent findings did suggest that pain at different body sites do share common genetic components. (ref 24-26).

Major concerns:

1. I would think that there are some study design issues in the first place. The authors used 4 musculoskeletal pain phenotypes in this study. Among these 4 pain phenotypes, in general, hip pain and knee pain are more closely related with OA (more likely caused by OA) than back pain and neck pain (OA is not the main reason for back pain and neck pain). So, the genetic mechanisms of hip pain and knee can be affected by the contributing genes of OA while those of back pain and neck pain will not get affected as much as hip pain and knee pain do. A better study design is to remove all the self-reported OA from the UK Biobank cohort. So the remaining case samples of 4 pain phenotypes are non-OA driven musculoskeletal pain phenotypes.

As it will be too laborious for re-analysing, so I will let it go. But as a reviewer, I need to point it out. (The authors did have some discussion about OA's impact to the GWAS results of GIPs and it is not surprising that some SNPs identified have connections to OA)

2. Among all eight pain phenotypes in the UK Biobank, Headache, facial pain and stomach pain are not in general considered as musculoskeletal pain. But why pain all over the body was not included? It is likely to be one of the musculoskeletal pain phenotypes and it might be more suitable to be included in the analysis than hip pain and knee pain.

At line 126, authors did explain that **pain all over the body was thought to have an underlying generalised propensity to pain**. I am not sure that this is correct.

Table below: The genetic correlations between pain all over the body and other 7 pain phenotypes from the UK Biobank. They indicated that the genetic mechanisms of pain all over the body were somehow similar as other pain phenotypes at different areas. Again, I am not asking for the pain all over the body to be included in the analysis as it will be too much to re-analysis.

3. The following query needs to be addressed clearly

	Headache	Facial pain	Neck or shoulder pain	Stomach or abdominal pain	Back pain	Hip pain	Knee pain
Pain all over body	0.62	0.43	0.79	0.73	0.69	0.81	0.36

**To
validat
e your
GIP
metho**

ds, you need to find out that using your methods, how GIPs work on other phenotypes. For example,

1. Pick 4 clearly genetically related phenotypes (like the four pain phenotypes you used) from the UK Biobank GWAS summary statistics (plenty of online resources to download), calculate GIP1 to GIP4 for these 4 phenotypes, then generate a Figure like Figure 2a and 2c and see what they show.
2. Pick 4 clearly genetically unrelated phenotypes from the UK biobank. Do the same as above. To see what they show.

Your GIP methods, if being correct, should work on 1, but if it also works on 2, then, the methods are not that useful.

Minor concerns:

1. Line 85, are you sure it is ref 18 not ref 20?
2. Line 101, 'we assume... through OA can unravel shared musculoskeletal pathways....'. In my personal opinion, the link of pain phenotypes to OA adds potential biases rather than considered as an advantage.
3. Line 130, why split the EU samples into 265,000 and 174,831? Any specific reasons for the discovery cohort to use 265,000 samples?
4. Line 140, you might want to replace the reference with the published one.
<https://www.nature.com/articles/s41586-018-0579-z>
Ref 25 has also been published. <https://www.ncbi.nlm.nih.gov/pubmed/31659249>
5. Line 264, the heritabilities from LD hub are always significantly lower than the GCTA calculated ones. Not so many researchers use those heritabilities generated by the LD hub. For example, the heritability of back pain in your manuscript is 0.04, but in the ref 20 (most authors are the same with this manuscript) is 0.08.
6. Figure 2d seems come out from nowhere. There are no texts in the manuscript describing how the phenotypic correlations were carried out.
7. Line 361, authors said that the SLC39A8 gene is associated with neck/shoulder pain, please indicate which paper?
8. Line 594, cite this <https://www.nature.com/articles/s42003-019-0568-2> or somewhere in the manuscript.
9. Line 649-651. Abdominal pain and headaches are clearly not related with OA, but why the rg values of the GIP1 for the 4 traits and for the 6 traits are so matched? Authors claim that GIP1 for musculoskeletal pain likely reflects chronic pain per se. That is a big claim. There are other reasons can explain the 99% similarity, such as authors are using the same cohort. If authors want to test that, **try GIP1 for 4 pain phenotypes in the UK Biobank and then use GIP1 for other pain phenotypes (not back, hip, knee and neck pain) from other cohorts as replications. That is more persuasive for that conclusion.**
10. Line 673. No, your phenotype was not similar with the one used in REF 108 at all. Ref 108 used numbers from 0 to 7 as phenotype values in the GWAS. But you used the resulting principal components (as you stated in the supplementary method).
I am one of the reviewers of ref 108. The paper does have other issues in addition to the one you mentioned. But, as I mentioned before, those attempts to identify genetic mechanisms of complex traits should be encouraged.

Reviewer #2 (Remarks to the Author):

This is a well-written and researched article with an important contribution to the field of chronic pain genetics. The authors are to be commended for the amount and quality of work put into it. The main contribution is a genetically derived chronic pain phenotype, which they convincingly offer as a construct that reduces heterogeneity in musculoskeletal chronic pain conditions. The new phenotype is obtained from principal component analysis of four persistent (3+ months duration) musculoskeletal conditions in different body sites, available in the UK Biobank. The selection of four pain phenotypes from the eight available in the UKB under Data Field 6159 is adequately explained. Together with the other three derived phenotypes from PCA, the first and most explanatory genetically independent phenotype (GIP) was used for a GWAS in the discovery and three replication samples, all taken from the UK Biobank. SNP effects, SNP heritability estimates (higher for the first GIP than for any of the individual four pain conditions), and genetic correlation with ~2300 additional traits were determined, and gene set and tissue/cell type enrichment analyses and pleiotropy analysis for overlapping effects on gene expression were carried out. Literature review was additionally done using Google Scholar, Pubmed, NCBI Gene, and OMIM to evaluate the biological relevance of genes implicated in their GWAS results to chronic musculoskeletal pain. The sum of these analyses and data compilations from previously published sources offers a thorough and comprehensive characterisation of the genetic component of chronic musculoskeletal pain given what is known to-date.

To the best of my knowledge, the descriptions of methods in the main text and supplementary methods would allow one to reproduce the work described and possibly apply the proposed GIP construction to another set of phenotypes with suspected common genotypic components.

The limitations are well discussed.

1. My only main concern is with a Caucasian discovery sample and two non-Caucasian validation samples, which individually do not have significant associations for the choice SNPs, Table S4. Why not instead split the UKB cohort into a discovery and replication samples using the same population, either all Caucasians or all participants, for example as 200K for discovery, then 2 or 3 sets of 100K (depending on whether the analysis is limited to Caucasians)? Another alternative, the two-step iterative resampling method, is evaluated in Kang et al. 2015 (PMID 26377241). Perhaps this would also get rid of the need for a meta analysis to show validation of discovery findings.

Small issues:

2. Figure S5 legend: "summary statistics were" should be "was", also on page 4 of Supp Methods

3. In Supplementary Methods:

a. Typo in the Phenotype Definition section "We defined choric pain patients" on p. 2

b. "To elucidate (the) genetic component" on p. 7

4. Typo in Table S12 "SNPs from the MHC (chr6 26M~34M) region was removed for this traits"

5. Table S7a lists genetic loci according to build GRCh38, but elsewhere GRCh37 is used. If possible, use the same build throughout for consistency.

Point-by-point response to the reviewers' comments:

Reviewer #1

Tsepilov et al has prepared a manuscript that aims to identify shared genetic components among chronic musculoskeletal pain occurred at different body sites. They adapted a new concept of genetically independently phenotypes (GIPs) to replace the traditional 'Yes or No' pain phenotypes at 4 body sites. It is a new attempt and should be encouraged since recent findings did suggest that pain at different body sites do share common genetic components. (ref 24-26).

Response: We would like to thank the Reviewer for the appreciative comments.

Reviewer #1

Major concerns:

1. I would think that there are some study design issues in the first place. The authors used 4 musculoskeletal pain phenotypes in this study. Among these 4 pain phenotypes, in general, hip pain and knee pain are more closely related with OA (more likely caused by OA) than back pain and neck pain (OA is not the main reason for back pain and neck pain). So, the genetic mechanisms of hip pain and knee can be affected by the contributing genes of OA while those of back pain and neck pain will not get affected as much as hip pain and knee pain do. A better study design is to remove all the self-reported OA from the UK Biobank cohort. So the remaining case samples of 4 pain phenotypes are non-OA driven musculoskeletal pain phenotypes.

As it will be too laborious for re-analysing, so I will let it go. But as a reviewer, I need to point it out. (The authors did have some discussion about OA's impact to the GWAS results of GIPs and it is not surprising that some SNPs identified have connections to OA).

Response: The Reviewer makes a good point about phenotype definition. We had given this a great deal of discussion among the group when we designed the study. Initially we had used more and diverse pain phenotypes, including headache and abdominal pain, but decided on balance that it was better to focus on the phenotypically-related musculoskeletal sites of hip, knee and upper and lower spine.

Self-reported OA as obtained from surveys or interviews tends to be a proxy for musculoskeletal pain, rather than accurately reflecting imaging-confirmed or surgically-confirmed osteoarthritis of synovial joints. This is because early OA on plain radiograph is often asymptomatic and surgical referral occurs when pain is chronic and not managed by analgesics. We agree with the Reviewer that there may be value in conducting secondary analyses removing subjects with radiographic osteoarthritis, but at the time we designed the study our main interest was in chronic pain mechanisms of whatever underlying cause. The problem with spine – and we have considerable combined experience in this field – is that imaging methods do not readily permit the combined assessment of disc degeneration and facet joint OA. Many argue that the imaging- and pathological features of intervertebral disc degeneration (matrix metalloproteinase activation, degradation of proteoglycans, loss of disc cells etc) is so very like that seen in OA it may be considered much the same process and phenotypically spine degeneration is correlated with peripheral joint OA, suggestive of similar age-related pathological change. Finally disc

degeneration has been shown in a number of studies in different settings to be the single biggest risk factor for episodes of back pain, so justified its inclusion.

We acknowledge that our electing to study chronic musculoskeletal pain phenotypes may have led to identification of some OA-related features (SNPs, genes, genetic correlations), and that we will see both general pain and musculoskeletal pathology components. For this reason, we have prominently discussed osteoarthritis as one possible component explaining our findings related to the GIPs, in discussion section paragraphs 6, 7, 8, and 11

Reviewer #1

2. Among all eight pain phenotypes in the UK Biobank, Headache, facial pain and stomach pain are not in general considered as musculoskeletal pain. But why pain all over the body was not included? It is likely to be one of the musculoskeletal pain phenotypes and it might be more suitable to be included in the analysis than hip pain and knee pain.

At line 126, authors did explain that pain all over the body was thought to have an underlying generalised propensity to pain. I am not sure that this is correct.

Table below: The genetic correlations between pain all over the body and other 7 pain phenotypes from the UK Biobank. They indicated that the genetic mechanisms of pain all over the body were somehow similar as other pain phenotypes at different areas. Again, I am not asking for the pain all over the body to be included in the analysis as it will be too much to re-analysis.

Response: Again, the Reviewer makes a very pertinent comment, and one which we had considered in designing the study.

Thank you for pointing out that the results in the Table indicate the genetic correlations are similar for pain all over as for other pain phenotypes. We have now modified the text in the Methods section and deleted the sentence about underlying generalised propensity to pain.

However, we had additional reasons not to include pain all over the body in a set of phenotypes for our analysis. First of all, this phenotype is *a priori* negatively correlated (phenotypically) with all other pain phenotypes (individuals who chose the answer “pain all over the body” could not select other pain sites in a web-based questionnaire). Second, pain all over the body has very low prevalence in the UK Biobank study sample (about 1.6%) leading to a low effective sample size and unstable heritability and genetic correlation estimates. Thus, we concluded that this phenotype would not be a good choice for our method.

Reviewer #1

3. The following query needs to be addressed clearly

To validate your GIP methods, you need to find out that using your methods, how GIPs work on other phenotypes. For example,

1. Pick 4 clearly genetically related phenotypes (like the four pain phenotypes you used) from the UK Biobank GWAS summary statistics (plenty of online resources to download), calculate GIP1 to GIP4 for these 4 phenotypes, then generate a Figure like Figure 2a and 2c and see what they show.

2. Pick 4 clearly genetically unrelated phenotypes from the UK biobank. Do the same as above. To see what they show.

Your GIP methods, if being correct, should work on 1, but if it also works on 2, then, the methods are not that useful.

Response: Thank you for your suggestion. We validated our approach using five scenarios. Two of these scenarios are extreme artificial cases (a set of completely genetically independent traits and a set of genetically identical traits), and three are real datasets.

Please find our results in Supplementary Methods, pp. 12-18.

In brief, the GIP1 captures large part of the variation for each trait in case of high genetic correlations between traits. If the traits are not genetically related, GIP1 does not approximate their shared genetic background.

Reviewer #1

Minor concerns:

1. Line 85, are you sure it is ref 18 not ref 20?

Response: Thank you for noticing this, we have amended the reference.

Reviewer #1

2. Line 101, ‘we assume... through OA can unravel shared musculoskeletal pathways....’ . In my personal opinion, the link of pain phenotypes to OA adds potential biases rather than considered as an advantage.

Response: Thank you. We have now corrected this sentence in order to avoid misunderstanding. We did not consider this as an advantage and realize that our phenotype choice (chronic site-specific musculoskeletal pain) may limit the generalizability of the results (please, see the limitations section at the end of the Discussion: “Second, in our study, we focused only on chronic musculoskeletal pain at anatomical sites potentially linked through osteoarthritis, so one must be cautious about generalizing our results to other chronic pain conditions”).

Reviewer #1

3. Line 130, why split the EU samples into 265,000 and 174,831? Any specific reasons for the discovery cohort to use 265,000 samples?

Response: We had decided *pre hoc* to split the EU sample by 265,000 and 174,834 in order to obtain the optimal balance of statistical power between discovery and replication. We estimated the statistical power for different sample sizes of discovery and replication cohorts (based on the effect sizes of the top SNPs from our previous back pain GWAS) and showed that splitting by $\frac{3}{4}$ for discovery and $\frac{1}{4}$ for replication would be an optimal ratio since it enabled reaching 80% power in both the discovery and the replication cohorts. We added this information in the Methods section (“Study sample and phenotype definition”).

Reviewer #1

4. Line 140, you might want to replace the reference with the published one.
<https://www.nature.com/articles/s41586-018-0579-z> Ref 25 has also been published.
<https://www.ncbi.nlm.nih.gov/pubmed/31659249>

Response: Thank you, we have amended these references.

Reviewer #1

5. Line 264, the heritabilities from LD hub are always significantly lower than the GCTA calculated ones. Not so many researchers use those heritabilities generated by the LD hub. For example, the heritability of back pain in your manuscript is 0.04, but in the ref 20 (most authors are the same with this manuscript) is 0.08.

Response: Thank you. In ref 20 (doi: 10.1371/journal.pgen.1007601), we used LD score regression to estimate the SNP-captured heritability of chronic back pain and reported heritability on the liability scale. In this manuscript, we presented heritability on the observed scale.

However, to address the reviewer's point, we now also provide heritability estimates on the liability scale (please, see Table S2a). For back pain, heritability on the liability scale calculated by LD score regression is 0.09, quite close to that reported in ref 20.

Additionally, we used REML algorithm implemented in BOLT-LMM v2.3.2 to calculate heritability on both the observed and the liability scale. The results can be found in Table S2a. All estimates were higher than those obtained using LD score regression.

Reviewer #1

6. Figure 2d seems come out from nowhere. There are no texts in the manuscript describing how the phenotypic correlations were carried out.

Response: Thank you for noting this. We updated our Supplementary Methods and added a reference to these data in the legend of Figure 2.

Reviewer #1

7. Line 361, authors said that the SLC39A8 gene is associated with neck/shoulder pain, please indicate which paper?

Response: Thank you. We meant the locus, not the gene. We corrected this sentence and other sentences in this paragraph which might cause confusion.

Reviewer #1

8. Line 594, cite this <https://www.nature.com/articles/s42003-019-0568-2> or somewhere in the manuscript.

Response: Thank you. We added the reference to Meng et al. study in the Results section (GWAS for genetically independent phenotypes, page 12, ref 56).

In our analysis we noted that the authors of this article mixed up the reference and effective alleles in Table 2 and in the text. For example, they indicate the allele rs143384 A (AF = 60.0%) as effective allele with beta -0.008 (UK Biobank). Thus, they state that this allele is a low-risk allele for knee pain.

In our study, the allele rs143384 A (AF = 59.8%) is positively associated with chronic knee pain (see Table S4: line 8, raw BY, discovery UK Biobank cohort, beta = 0.023; line 8, raw CK, UK Biobank European meta-analysis, beta = 0.0217). Thus, in our work the allele rs143384 A is a high-risk allele for pain at this location.

Allele rs143384 A is positively correlated with allele rs143383 A ($r^2 = 0.82$ in Eur): <https://ldlink.nci.nih.gov/?var1=rs143384&var2=rs143383&pop=CEU%2BTSI%2BFIN%2BGBR%2BIBS&tab=ldpair>, which corresponds to a positive correlation between the negative strand alleles rs143384 T and rs143383 T. Allele rs143383 T is a well-known risk allele for OA (please, see doi:10.1093/hmg/ddm174 and other refs in our Table S5). It is also associated with congenital dislocation of the hip and lumbar disc degeneration. Thus, a high-risk chronic knee pain allele (as indicated in our study) is linked with a high-risk OA allele.

We checked the Data availability link in the study by Meng et al.

<https://figshare.com/articles/kneepaingwas/9611198>

and found out that allele rs143384 A is a high-risk allele – as in our study. So, the mistake occurred at the stage of interpretation, not calculations. Other alleles in Meng et al.'s Table 2 have the same problem.

We have written to Meng et al. to notify them of these observations.

Reviewer #1

9. Line 649-651. Abdominal pain and headaches are clearly not related with OA, but why the rg values of the GIP1 for the 4 traits and for the 6 traits are so matched? Authors claim that GIP1 for musculoskeletal pain likely reflects chronic pain per se. That is a big claim. There are other reasons can explain the 99% similarity, such as authors are using the same cohort. If authors want to test that, try GIP1 for 4 pain phenotypes in the UK Biobank and then use GIP1 for other pain phenotypes (not back, hip, knee and neck pain) from other cohorts as replications. That is more persuasive for that conclusion.

Response: Thank you. We agree with you that there can be other reasons explaining the similarity. We have now made changes to soften our statements.

Unfortunately, we were not able to apply our GIP method using independent cohorts, because those cohorts to which we have access do not have enough effective sample size for the traits of interest to be included in the analysis. However, we generated polygenic risk scores using our GIP1 GWAS hits ($P < 1.0e-5$) and analysed their association between pain-related traits in the TwinsUK cohort.

The results are given in the Table below. We observed associations ($P < 0.05$) with 2 musculoskeletal pain conditions and pelvic pain, dry eye syndrome (which is characterized by

pain in the eyes), and irritable bowel syndrome (which is characterized by abdominal pain; however, we did not find any association with abdominal pain as a separate phenotype).

Association between PRS generated using the GIP1-associated SNPs with $P < 1.0e-5$ and pain-related phenotypes in the TwinsUK cohort.

Trait (sample size)	Estimate	SE	z	Pr(> z)
neck pain (N = 3,130)	0.07	0.04	1.61	0.11
chronic widespread pain (N = 3,273)	0.33	0.14	2.43	0.02
low back pain (N = 4,190)	0.17	0.04	3.86	0.0001
dry eye syndrome* (N = 3,456)	0.94	0.41	2.31	0.02
pelvic pain (N = 3,541)	0.85	0.38	2.24	0.02
irritable bowel syndrome** (N = 4,169)	1.02	0.44	2.32	0.02
headache (N = 4,150)	1.82	2.79	0.65	0.51
sternum pain (N = 4,149)	1.10	2.25	0.49	0.62
abdomen pain (N = 4,150)	-0.71	1.70	-0.42	0.68

* characterized by pain in the eyes

** characterized by abdominal pain

Reviewer #1

10. Line 673. No, your phenotype was not similar with the one used in REF 108 at all. Ref 108 used numbers from 0 to 7 as phenotype values in the GWAS. But you used the resulting principal components (as you stated in the supplementary method). I am one of the reviewers of ref 108. The paper does have other issues in addition to the one you mentioned. But, as I mentioned before, those attempts to identify genetic mechanisms of complex traits should be encouraged.

Response: Thank you. We agree with you that phenotypes are not similar. We corrected this part of the text.

Reviewer #2

This is a well-written and researched article with an important contribution to the field of chronic pain genetics. The authors are to be commended for the amount and quality of work put into it. The main contribution is a genetically derived chronic pain phenotype, which they convincingly offer as a construct that reduces heterogeneity in musculoskeletal chronic pain conditions. The new phenotype is obtained from principal component analysis of four persistent (3+ months duration) musculoskeletal conditions in different body sites, available in the UK Biobank. The selection of four pain phenotypes from the eight available in the UKB under Data Field 6159 is adequately explained. Together with the other three derived phenotypes from PCA, the first and

most explanatory genetically independent phenotype (GIP) was used for a GWAS in the discovery and three replication samples, all taken from the UK Biobank. SNP effects, SNP heritability estimates (higher for the first GIP than for any of the individual four pain conditions), and genetic correlation with ~2300 additional traits were determined, and gene set and tissue/cell type enrichment analyses and pleiotropy analysis for overlapping effects on gene expression were carried out. Literature review was additionally done using Google Scholar, Pubmed, NCBI Gene, and OMIM to evaluate the biological relevance of genes implicated in their GWAS results to chronic musculoskeletal pain. The sum of these analyses and data compilations from previously published sources offers a thorough and comprehensive characterisation of the genetic component of chronic musculoskeletal pain given what is known to-date.

To the best of my knowledge, the descriptions of methods in the main text and supplementary methods would allow one to reproduce the work described and possibly apply the proposed GIP construction to another set of phenotypes with suspected common genotypic components.

The limitations are well discussed.

Response: Thank you for your encouraging comments.

Reviewer #2

1) My only main concern is with a Caucasian discovery sample and two non-Caucasian validation samples, which individually do not have significant associations for the choice SNPs, Table S4. Why not instead split the UKB cohort into a discovery and replication samples using the same population, either all Caucasians or all participants, for example as 200K for discovery, then 2 or 3 sets of 100K (depending on whether the analysis is limited to Caucasians)? Another alternative, the two-step iterative resampling method, is evaluated in Kang et al. 2015 (PMID 26377241). Perhaps this would also get rid of the need for a meta analysis to show validation of discovery findings.

Response: Thank you for raising this interesting point. We had long discussions about the best approach to adopt, within the group, before alighting on the method presented in the manuscript. We decided to a ‘canonical’ replication method that assumes validation of the top SNPs identified in discovery GWAS in independent samples). Using this design, we guarantee, at least in theory, that the type 1 error for our replicated findings is less than 0.05^2 (0.0025). Such small type 1 error is essential for our extensive in-silico follow up analysis. Also, our intention was to reach the greatest power in discovery GWAS while keeping the power of replication not less than 80%. We did not restrict the replication cohort to Caucasians only aiming to increase the available sample size to the maximum.

The Reviewer raises an interesting alternative in the 2-step iterative resampling approach (TSIR). We adopted this approach to SNPs associated with GIP1 at a genome-wide level on the joint meta-analysis of all European-ancestry samples (59 SNPs on the total sample size of 439K). We used the following settings described in the paper suggested by the Reviewer (PMID: 26377241): $\pi = 0.8$; $\alpha_1 = 1e-5/(\text{Number of SNPs})$; $\alpha_2 = 0.05/(\text{Number of SNPs})$; number of iterations = 100; r was set to 20. Using this approach, 7 loci were considered replicated by TSIR. Three out of 7 overlapped with our 5 replicated SNPs. Since the results and the power of TSIR analysis were close to the canonical approach, we decided to stick to the canonical one.

Reviewer #2

2) Small issues:

2. Figure S5 legend: "summary statistics were" should be "was", also on page 4 of Supp Methods

Response: Thank you. We double-checked the text and corrected this issue. Now “summary statistics” is followed by “is” or “was” (singular).

Reviewer #2

3. In Supplementary Methods:

- a. Typo in the Phenotype Definition section "We defined choric pain patients" on p. 2
- b. "To elucidate (the) genetic component" on p. 7

Response: Thank you. We corrected these typographical errors.

Reviewer #2

4. Typo in Table S12 "SNPs from the MHC (chr6 26M~34M) region was removed for this traits"

Response: Thank you. We corrected this error.

Reviewer #2

5. Table S7a lists genetic loci according to build GRCh38, but elsewhere GRCh37 is used. If possible, use the same build throughout for consistency.

Response: Thank you. We corrected the legend in Table 7a (it is the GRCh37 build, but we erroneously indicated it as GRCh38).

REVIEWERS' COMMENTS:

Reviewer #1 (Remarks to the Author):

The authors have addressed my queries sufficiently. I can see that they worked hard for the revision. Many thanks for the extra analysing work to validate the GIP method using multiple other datasets (Figure SM1-5).

I noticed that authors only share their GWAS summary statistics upon reasonable request. This seems not following the journal's requirement. But it will be the chief editor's call.

Reviewer #2 (Remarks to the Author):

In my opinion, the issues brought up by both reviewers have been adequately addressed and resolved, when feasible.

Also, I owe the authors an apology for my previous comment about summary statistics needing to be treated as a singular noun. In fact, when referring to statistics as pieces of data rather than statistics as the discipline, it should be plural. So please change the sentences with mention of "summary statistics" back to the plural verb. Sorry about the unnecessary confusion.

Point-by-point response to the reviewers' comments:

Reviewer #1 (Remarks to the Author):

The authors have addressed my queries sufficiently. I can see that they worked hard for the revision. Many thanks for the extra analysing work to validate the GIP method using multiple other datasets (Figure SM1-5).

Response: Many thanks for this encouraging comment.

Reviewer #1

I noticed that authors only share their GWAS summary statistics upon reasonable request. This seems not following the journal's requirement. But it will be the chief editor's call.

Response: Now we have made GWAS summary statistics publicly available (please, see the Data availability statement; Zenodo: <http://dx.doi.org/10.5281/zenodo.3797553>).

Reviewer #2 (Remarks to the Author):

In my opinion, the issues brought up by both reviewers have been adequately addressed and resolved, when feasible.

Response: Thank you.

Reviewer #2

Also, I owe the authors an apology for my previous comment about summary statistics needing to be treated as a singular noun. In fact, when referring to statistics as pieces of data rather than statistics as the discipline, it should be plural. So please change the sentences with mention of "summary statistics" back to the plural verb. Sorry about the unnecessary confusion.

Response: Thank you. We corrected this issue throughout the manuscript body and Supplementary materials.